# A Moderated-Mediation Model of the Relationship between Dietary Satisfaction and Fatigue in Older Adults with Diabetes: The Role of Meal Planning and Depressive Symptoms

**DOI:** 10.3390/ijerph17238823

**Published:** 2020-11-27

**Authors:** Hyerang Kim, Heesook Son

**Affiliations:** 1Department of Nursing Science, Howon University, 64 Howondae 3gil, Impi, Gunsan 54058, Jeollabuk-do, Korea; hkim167@hotmail.com; 2Red Cross College of Nursing, Chung-Ang University, 84 Heukseok-ro, Dongjak-gu, Seoul 06974, Korea

**Keywords:** diabetes, diet-related quality of life, difficulty with meal planning, depressive symptoms, fatigue, older adults, satisfaction with diet

## Abstract

Little research has examined the pathways between psychological factors and fatigue in older adults with diabetes. This study explored the pathways between diet-related quality of life and depressive symptoms in predicting fatigue using a moderated-mediation model. A convenience sample of adults ≥65 years (*n* = 127) with diabetes completed a cross-sectional survey including measures of fatigue severity, diet-related quality of life, and depressive symptoms, and a moderated-mediation analysis assessed the relationships between them. Diet satisfaction was negatively related to fatigue, which was mediated by depressive symptoms. In the moderated-mediation model, diet satisfaction had a conditional effect on fatigue through the mediating effect of depressive symptoms, moderated by meal planning difficulty. At higher levels of perceived meal planning difficulty, lower diet satisfaction was indirectly associated with higher fatigue through depressive symptoms, but this pathway was non-significant at lower levels. Findings suggest that supportive care for diet therapy might improve psychological outcomes in older adults with diabetes, especially for those having difficulties with daily dietary practice. Meal planning difficulties in the dietary management of diabetes accompanied by low diet satisfaction may lead to negative psychological outcomes. Monitoring satisfaction and burdens associated with dietary practices could improve fatigue in this population.

## 1. Introduction

The global prevalence of diabetes in adults aged 65 years or older is approximately 20% due to the growth in the aging population and extended life expectancy [1]. The progression of diabetes in older adults differs from other age groups since the double burden of aging and diabetes synergistically imposes physical, psychological, and social impairment, increasing the risk of diabetes complications and mortality [2].

Fatigue, the second most commonly reported symptom of diabetes, is defined as “a subjective lack of physical and/or mental energy that is perceived by the individual or caregiver to interfere with usual or desired activities” [3,4,5]. Diabetes fatigue has been reported to be influenced by multiple factors, including physiological and psychological status and lifestyle [6,7,8]. These factors can have a cyclic relationship with diabetes fatigue, such that they negatively affect optimal blood glucose control, which, in turn, increases the risk of undesirable health outcomes [8,9]. Older adults with diabetes have been shown to be more vulnerable to fatigue than older adults without diabetes and younger adults with diabetes [10,11,12] since both age- and diabetes-related symptoms synergistically contribute to an increased risk for fatigue in older adults [3,13,14]. Despite these risks for fatigue in older adults with diabetes, little is known about the factors contributing to fatigue in this population.

Diet-related quality of life generally refers to the impact of dietary practices and diet-related psychological outcomes on a person’s quality of life [15] and has been evaluated by measuring diet-related psychological factors, including diet satisfaction, burdens of dietary restriction, the perceived value of diet therapy, and difficulties with meal planning [16]. Older adults face challenges when achieving an optimal diet-related quality of life since physical, psychological, and cognitive declines may adversely influence daily engagement with meal planning and dietary practices along with diet satisfaction [17,18,19,20]. Poor diet-related quality of life in older adults could lead to physical and psychological exhaustion and fatigue, which, in turn, interfere with usual or desirable dietary practices and optimal quality of life [11,13]. While diet-related quality of life may potentially influence fatigue in older adults, it has rarely been examined as a predictor of fatigue in older adults with diabetes [8,21]. Previous research on diabetes fatigue has primarily focused on dietary intake, eating behaviors, anthropometric measures of nutritional status, and nutrition-related health outcomes [8,9]. Given the potential reciprocal relationships of the components of diet-related quality of life, further research is needed to address the complex pathways between these variables and the dynamics between them in predicting fatigue in older adults with diabetes.

Fatigue is also a common symptom of depression [8,9,22]. Older adults with diabetes are at increased risk for co-occurring depressive symptoms and fatigue [11,13,23,24,25], and the co-occurrence of depressive symptoms and fatigue increases threefold in older adults with diabetes [26]. The presence of both fatigue and depressive symptoms in older adults with diabetes has been significantly associated with age-related physical and psychological changes accompanied by poor functional performance, higher levels of disability, and a suboptimal lifestyle [27]. Further, diet-related quality of life has been found to be closely associated with mental health, including depressive symptoms, in older adults compared to other age groups, indicating that diet acts as a means of food provision but a crucial aspect of one’s social and psychological life [28]. The burdens and challenges associated with the responsibilities of daily dietary self-management also contribute to an increased risk for depressive symptoms, which could contribute to fatigue [29,30,31,32]. However, little research has examined the complex relationship between diet-related psychological factors, depressive symptoms, and diabetes fatigue. The close association between diet-related quality of life and depressive symptoms and its influence on fatigue has been established. However, the relationship between diet-related quality of life and fatigue in the presence of co-occurring depressive symptoms, particularly with depressive symptoms mediating the relationship between diet-related psychological factors and fatigue, has not been examined. Clarifying the relationships between these psychological factors could help develop targeted interventions to address diet-related quality of life among older adults with diabetes at risk for fatigue.

Therefore, this study aimed to examine the complex pathways between diet-related quality of life, depressive symptoms, and fatigue. More specifically, an analysis was conducted to examine the interrelationship between diet-related quality of life and fatigue severity and the mediating role of depressive symptoms in the relationship between factors of diet-related quality of life and fatigue.

## 2. Materials and Methods

### 2.1. Participants

Using a cross-sectional design, the participants were recruited from a community health center and a seniors’ center using convenient sampling methods. The inclusion criteria for participants included (1) having a diagnosis of diabetes and currently taking antihyperglycemic agents, (2) being aged 65 years or older, and (3) living in their own home. Potential participants were given study information, including details regarding the study purpose, procedure, the right to withdrawal, confidentiality obligations, and the voluntary nature of their participation in the study. A prescreening interview was conducted to exclude any individuals with acute or malignant conditions that might be related to fatigue, including (1) terminal diseases with life expectancy less than 6 months and (2) hospitalizations or emergency room visits at least twice due to acute hyperglycemic or hypoglycemic events and other acute inflammatory diseases in the 3 months prior to the study. This study was approved by the institutional research board (IRB) where the corresponding author affiliated (IRB No. 1041078-201901-HR-003-01). The face-to-face questionnaire survey was collected from March to May in 2019.

### 2.2. Variables

Sociodemographic (i.e., age, gender, marital status, education, and household income) and clinical (i.e., years of having diabetes and number of comorbidities) characteristics, and eating context (i.e., frequency of exclusive eating alone and eating out) were collected using a structured questionnaire. Psychological characteristics, including fatigue, factors of diet-related quality of life, and depressive symptoms, were measured using the instruments indicated below.

Fatigue was measured using the Fatigue Severity Scale (FSS), developed by Krupp and colleagues [33]. The scale consists of nine questions asking the extent of fatigue during the past week on a 7-point rating scale (1 = *not at all* to 7 = *extremely severe*). The FSS has been used to assess chronic fatigue that is commonly associated with chronic disease, as it focuses on the impact of fatigue in daily life with regards to the accumulating effects of functional fatigue [34]. The total mean score was calculated with higher scores indicating more severe fatigue. The cutoff values included less than 4 points for normal, 4–4.9 for moderate fatigue, and 5 or more for severe fatigue. The Cronbach’s alpha coefficients were 0.93 in the original study and 0.92 in this study.

Factors of diet-related quality of life included satisfaction with diet, burden of diet therapy, perceived merits of diet therapy, and difficulty with meal planning. The 3 variables of satisfaction with diet, burden of diet therapy, and perceived merits of diet therapy were measured using the Diabetes Diet-Related Quality of Life–Revised version (DDRQOL-R-9) developed by Sato and colleagues [15]. It contains nine items on a 7-point Likert type scale ranged from 1 (very strongly disagree) to 7 (very strongly agree) in three subscales: “satisfaction with diet” (3 items), “burden of diet therapy” (3 items), and “perceived merits of diet therapy” (3 items). The sum of the scores for each scale was evaluated in total scores of 100 points, with higher scores for each subscale indicating better “satisfaction with diet,” “perceived merits of diet therapy,” and higher “burden of diet therapy.” The DDRQOL-R-9 demonstrated good internal consistency with Cronbach’s alpha coefficients of 0.86 for satisfaction with diet, 0.86 for burden of diet therapy, and 0.82 for perceived merits of diet therapy in the original study. It was translated into Korean by a bilingual nursing researcher through translation and back-translation procedure. The internal consistency in this study was satisfactory (Cronbach’s alpha coefficient = 0.90 for satisfaction with diet, 0.87 for burden of diet therapy, and 0.80). A question regarding the difficulty with meal planning was added to assess the psychological burden of putting meal planning into daily dietary practices, using a 5-point Likert scale (1 = strongly disagree to 5 = strongly agree).

Depressive symptoms were measured using the Geriatric Depression Scale-15 (GDS-15) developed by Sheikh and Yesavage [35] and translated into Korean by Jang and colleagues [36]. The scale is a self-administered instrument to evaluate depressive symptoms in older adults. It consists of 15 questions rated with yes or no based on how the respondent has felt over the past week. Out of a total score of 15 points, 0–5 points were considered not-depressed, 6–8 points as mild depression, and ≥9 points as severe depression. The internal consistency of the Kuder–Richardson Formula 20 (KR-20) was 0.92 in this study.

### 2.3. Statistical Analyses

Univariate analyses using independent *t*-tests and one-way ANOVAs were used to examine sociodemographic, clinical, and psychological characteristics and eating context by the prevalence and severity of fatigue. We conducted conditional process analysis to test the interactive relationship between the test variables, including factors for diet-related quality of life (e.g., diet satisfaction, burden of diet therapy, perceived merits of diet therapy, and difficulty with meal planning), depressive symptoms, and fatigue, and the interaction of diet-related quality of life and depressive symptoms on fatigue in two interlinked steps. First, we conducted a hierarchical regression analysis to test a simple mediation model using a bootstrapping method. The unstandardized *b* value was used to assess how strongly each predictor influenced fatigue. Bootstrapping analysis of the regression models was conducted to estimate the bias-corrected and accelerated (BCa) confidence intervals (CI) with 1000 replications. These values were used to determine the 95% CI for each variable. An effect was considered significant when the BCa CI did not include a zero. The adjusted *R*^2^ was used to assess the variance in the domain score explained by the model.

Next, we empirically tested the overall moderated-mediation model (Model 7) using the PROCESS syntax procedure version 3.3 for SPSS (Guilford Publications: New York, NY, USA) [37]. We examined the strength of the direct and indirect effects and conditional moderating effects of predictor variables on fatigue through these procedures. Regression coefficients (*b*) and bias-corrected 95% CIs for *b* were calculated using a bootstrapping procedure (5000 samples). The direction of the relationship between the predictors and outcome variable through a conditional moderator was examined to interpret the moderating effects using a simple slope analysis. All statistical analyses were conducted using IBM SPSS Statistics 25 (IBM Corp., New York, NY, USA). Statistical significance was determined at *p* < 0.05.

## 3. Results

Table 1 shows the prevalence and severity of fatigue by participants’ characteristics. No differences in the prevalence and severity of fatigue were found for the sociodemographic and clinical characteristics; however, significant differences were observed for psychological factors and eating context. Specifically, the prevalence and severity differed by the difficulty with meal planning (*p* = 0.014) and depressive symptoms (*p* < 0.001) from the psychological factors and exclusively eating alone (*p* = 0.037) and the frequency of eating out (*p* = 0.028) from the eating context variables. The severity of fatigue was higher in those who were less satisfied with their diet (*p* = 0.004), reported greater difficulty with meal planning (*p* = 0.012), and had severe depression (*p* < 0.001). Fatigue was also higher in those who ate exclusively alone (*p* = 0.037) and ate out less frequently (*p* = 0.028).

Table 2 presents the regression model that tested the mediation effect of depressive symptoms on fatigue. Depressive symptoms were significantly associated with three variables of diet satisfaction (*b* = −0.292, 95% CI [−0.453, −0.130]), perceived value of diet therapy (*b* = −0.241, 95% CI [−0.409, −0.072]) and difficulty with meal planning (*b* = 0.487, 95% CI [0.118, 0.855]), but not with the burden of dietary restriction and eating context (adjusted *R^2^* = 0.197, *p* < 0.001). Fatigue severity was significantly related to diet satisfaction (*b* = −0.155, 95% CI [−0.239, −0.071]) and difficulty with meal planning (*b* = 0.328, 95% CI [0.136, 0.519; Model 1: adjusted *R*^2^ = 0.164, *p* < 0.001). After depressive symptoms were taken into account (Model 2), depressive symptoms were shown to be positively related to fatigue (*B* = 0.197, 95% CI [0.110, 0.284]), and the effects of diet satisfaction (*b* = −0.098, 95% CI [−0.180, −0.016]) and difficulty with meal planning (*b* = 0.232, 95% CI [0.049, 0.415]) on fatigue became weaker, but remained significant, increasing the model’s explanatory power by about 11% (Model 2: adjusted *R*^2^ = 0.279, *p* < 0.001).

We tested three additional conditions to determine the moderated-mediation relationships of the predictor variables on fatigue. First, we determined if there was a significant interaction between satisfaction with diet and difficulty with meal planning on depressive symptoms (Condition 1). Then, we tested for a different conditional indirect effect of satisfaction with diet on fatigue via depressive symptoms (Condition 2). Finally, we tested if there was a different conditional indirect effect of difficulty with meal planning on fatigue via depressive symptoms (Condition 3). The results showed that the interaction between diet satisfaction and difficulty with meal planning significantly predicted depressive symptoms (*b* = −0.107, 95% CI [−0.211, −0.004]), which supported Condition 1 (Table 3). A simple slope analysis visualized the relationship between diet satisfaction and depressive symptoms at three different levels of difficulty with meal planning (i.e., low [−1 SD], mean, high [+1 SD]; Figure 1). The relationship between diet satisfaction and depressive symptoms was greater at the mean (*b* = −0.316, 95% CI [−0.481, −0.150]) and a high level of difficulty with meal planning (*b* = −0.484, 95% CI [−0.725, −0.244]) but was not significant at a low level of difficulty with meal planning. The conditional indirect effect of diet satisfaction on fatigue differed across the level of difficulty with meal planning. It was significant at the mean (*b* = −0.070, 95% CI [−0.114, −0.032]) and a high level of difficulty with meal planning (*b* = −0.107, 95% CI [−0.173, −0.042]) but was not significant at a low level of difficulty with meal planning (Condition 2). In contrast, the conditional indirect effect of difficulty with meal planning was not found (Condition 3). Thus, taken together, the results supported moderated-mediation effects of difficulty with meal planning on the relationship between diet satisfaction and fatigue, mediated through depressive symptoms (Figure 2).

## 4. Discussion

It has been acknowledged that diet-related quality of life and depressive symptoms and their influence on fatigue are closely related, but little is known about the specific mechanism by which these factors lead to fatigue among older adults with diabetes. Our results showed that satisfaction with diet and difficulty with meal planning from the measured aspects of diet-related quality of life were significantly associated with fatigue. Our results provide additional information on the mechanism by which satisfaction with diet, difficulty with meal planning, and depressive symptoms contribute to fatigue, demonstrating that depressive symptoms mediated the relationship between diet satisfaction and fatigue. The mediating effects of depressive symptoms differed by the level of difficulty with meal planning. Individuals who had increased difficulty with meal planning were at increased likelihood of being dissatisfied with one’s current diet, a greater risk for depressive symptoms and fatigue. Meanwhile, the link between satisfaction with diet and depressive symptoms or fatigue for those having low difficulty with meal planning was not significant.

The most challenging part of diabetes management is patient adherence to diabetes dietary recommendations [38]. Since no standard meal plan or eating pattern universally works for all people with diabetes, adults with diabetes face challenges with their daily dietary practice, particularly decisions about what and how much to eat and implementing their meal planning [39]. Meal planning is a goal-directed behavior that requires complex cognitive abilities, including executive functioning, attention, and memory, for shopping, preparing, and cooking a meal [40]. Diminished capacity for daily self-care could increase the burdens of dietary practice, including meal planning among older adults with diabetes. Multiple factors related to both aging and diabetes could diminish older adults’ capacity for dietary self-management, which, in turn, could affect dietary adherence, motivation, attitude, and self-efficacy for meal planning, and actual engagement in dietary practices [41].

Compared to younger adults with diabetes, older adults may have increased difficulty with meal planning for various reasons, including functional decline, lack of social support, and economic constraints [42]. For example, impaired self-care capacity, along with functional decline, can impose physical and psychological burdens on older adults that affect daily tasks related to dietary practices, such as hindering one’s ability to prepare, cook, and eat food [43]. Indeed, patients with diabetes who did not perceive self-management practices as a burden were more likely to engage in these practices, which, in turn, led to an increased likelihood of following dietary guidelines leading to reduced fatigue [44]. Research has found that social isolation and the lack of social support are associated with food insecurity and an unhealthy diet, increasing nutritional risks [42,43,45].

Older adults are particularly vulnerable to economic security, influencing their ability to afford nutrient-rich foods that promote health and assist with disease management when making food choices [41]. A higher perceived burden of dietary self-management is associated with increased fatigue, depressive symptoms, and diabetic symptoms in patients with diabetes [46]. Nanayakkara and colleagues [47] reported that greater difficulty with following dietary recommendations was associated with increased depressive symptoms and diabetes distress among adults with diabetes. Engaging in dietary practice is essential for feeding oneself, increasing self-esteem and autonomy, and maintaining social roles. Thus, limited engagement in dietary practices could challenge older adults’ confidence in their ability to maintain a healthy, balanced diet, which, in turn, could increase the burden and distress of adhering to a diabetes diet-regimen [48,49,50].

Ahlgren and colleagues [51] reported that diet satisfaction was higher when adults with diabetes are more actively engaged in meal planning practices guided by a registered dietitian. Another study demonstrated that social support could reinforce diabetes self-care management, including engaging in daily dietary practices, participating in regular exercise, and following a medication schedule [52]. Given the influence of difficulty with meal planning as a moderating influence on depressive symptoms in the relationship between diet satisfaction and fatigue in our study, we identified a need for psychological interventions for older adults with diabetes that provide supportive care for dietary self-management. Interventions that help with decision-making regarding what and how much to eat to maintain optimal blood glucose levels and prevent diabetic complications are particularly needed. Such programs might mitigate the risk of depressive symptoms or fatigue among those with both low diet satisfaction and high perceived psychological burden of dietary self-management.

Satisfaction with diet generally changes over time, along with degenerative aging-related functional changes in daily dietary practices [53]. Further, diet satisfaction has been found to be significantly lower in older adults than younger adults, which is associated with diminished appetites and impaired physical abilities to perform dietary practice [50,54,55]. Having diabetes as an older adult may lead to unavoidable dietary modification or restrictions, which, when resulting in an unacceptable or unpalatable diet, may challenge older adults’ life-long food preferences and dietary habits [56,57]. These changes could contribute to poor diet-related quality of life with compromised enjoyment and diet satisfaction, resulting in depressive symptoms or emotional distress [58].

Our study demonstrated that participants reporting low diet satisfaction were more likely to have their fatigue mediated by depressive symptoms. The partial mediating effect of depressive symptoms in the relationship between diet satisfaction and fatigue suggests that depressive symptoms may contribute to developing fatigue in older adults with diabetes, indicating that diet dissatisfaction is positively associated with negative psychological well-being, leading to increased fatigue. As depressive symptoms appeared to be a factor increasing the prevalence and severity of fatigue, prolonged diet dissatisfaction might contribute to developing fatigue via the associated negative psychological conditions, like depressive symptoms. The co-occurrence of depressive symptoms and fatigue has been well-established in previous diabetes studies; however, research on the causal relationship between these variables has been limited [13,23,24,25]. Our study is the first to identify a relationship between fatigue and depressive symptoms in the context of diet-related quality of life. Dietary intervention for diabetes in older adults should consider the psychological value of diet in patients’ lives and the benefits of additional psychological interventions that focus on strategies to ensure diet satisfaction, which might improve older adults’ psychological outcomes and their increased adherence to dietary self-management for diabetes.

Our findings suggest several implications for future research and clinical practice. First, we included diet-related psychological factors as predictors for diabetes fatigue. Diet-related factors that influence fatigue have been previously addressed in the conceptualization of diabetes fatigue. Kalra and Sahay [8] viewed diabetes fatigue as a multifactorial syndrome caused by various lifestyle, diet-related, medical, psychological, and diabetic metabolism-related factors. However, their study focused primarily on dietary intake and anthropometric measures of nutritional status with less emphasis on the psychological aspects of diet-related issues. Similarly, both Fritschi and Quinn [6] and Griggs and Morris [9] suggested that lifestyle factors and diabetes self-management were associated with fatigue in diabetes. However, their conceptual framework did not clarify diet-specific lifestyle patterns and self-management practices, such as fatigue-related factors. Furthermore, they did not specify diet-related psychological issues. The current study is the first try to explain the association between diet-related psychological factors and fatigue.

Second, we used multiple variables related to diet-related quality of life and examined their interrelationship with fatigue to provide a detailed explanation of the mechanism leading to fatigue in older adults with diabetes. In our study, the diet-related quality of life variables were expanded to include both positive aspects, such as diet satisfaction and perceived value of diet, and negative aspects, including difficulty with meal planning and the burden of dietary restrictions. We found that fatigue was directly associated with diet satisfaction and indirectly related to difficulty with meal planning, and these associations were mediated by depressive symptoms. Further, the complex relationship between psychological factors, including diet-specific and other psychological factors, may affect fatigue. However, our study included a limited number of psychological factors and did not assess social support and other contextual factors, such as physical and social functioning and the environment for dietary practices, which may significantly influence psychological characteristics. Further research that designs a conceptual model with more indicators would help understand diabetes fatigue and suggest a strategic approach for diabetes intervention that might be particularly effective for populations vulnerable to fatigue.

Third, in terms of clinical implication, the study suggested areas to target for intervention to mitigate the risk of negative psychological outcomes, such as depressive symptoms and fatigue. Difficulty with meal planning, one of the major barriers to adherence to dietary self-management [40], was shown to be a moderating factor in the relationship between diet satisfaction and fatigue, suggesting the need to address difficulties with meal planning. However, in our current study, we did not include detailed information regarding the type or extent of difficulty with meal planning, nor did we assess what participants needed to learn or the type of help they needed. Research using qualitative and quantitative assessment of these aspects of meal planning is needed to develop nutrition interventions that address meal planning difficulties. These nutritional interventions should be optimized by considering patients’ physical, psychological, social, and cognitive functional status and the state of their degenerative aging process.

There are some limitations in the current study due to its cross-sectional design and relatively small sample size. First, although we included multiple predictors (i.e., sociodemographic, clinical, psychological, and contextual factors) to comprehensively examine the multidimensional characteristics of fatigue, the number of variables that were used was limited regarding the information related to diabetes fatigue. Our study did not include diabetes-specific clinical indicators, such as current hemoglobin A1c, variability of blood glucose level, and diabetes symptoms. Other factors not specific to diabetes, for example, medications potentially affecting fatigue and/or depressive symptoms, physical functions affecting either daily dietary practices (e.g., vision, hearing, cognitive function, frailty, and mobility) or dietary intake (e.g., difficulties in chewing and swallowing), and treatment burdens for diabetes or other comorbid conditions (e.g., taking several different medications) were not included in this study. Thus, it could not examine the influence of these factors on diet-related quality of life, depression, fatigue, or their association. The included diet-related factors were also limited to the psychological aspects and eating context and did not include actual dietary intake, eating patterns, nutrition literacy, and adherence to a dietary regimen. Our sample was limited to older adults with diabetes; thus, our results cannot be generalized to populations of other age groups with diabetes or healthy older adults without diabetes to determine the extent of diet-related quality of life, the severity of depressive symptoms and fatigue, and the dynamics of these variables in different population subgroups. Further, cross-sectional studies with larger samples of distinct subgroups would help assess the generalizability of our findings. Longitudinal observational studies and clinical trials are required to determine the causality of the relationships in this study. We also recommend the inclusion of living situation (e.g., living alone or with others) and eating context (e.g., eating alone or with others) in future research, as these variables could influence general and diet-related psychological health.

## 5. Conclusions

The current study examined the complex relationship between diet-related quality of life and depressive symptoms in predicting fatigue in a sample of older adults with diabetes. The findings supported the theory that fatigue was associated with decreased diet satisfaction, which was influenced by difficulties with meal planning in older adults with diabetes. Dissatisfaction with diet-related life and the psychological burdens of dietary management in older adults with diabetes appear to contribute to increased psychological distress such as depressive symptoms. When these negative emotions are accompanied by a suboptimal diet-related quality of life, they can continue or worsen, which may hamper the adoption or adherence to dietary self-management regimens. These findings provide additional information on the mechanisms of the relationship between diet-related psychological factors, depressive symptoms, and fatigue in older adults with diabetes. Specifically, the conditional indirect relationship between satisfaction with diet and fatigue with the moderated-mediation effect of difficulty with meal planning through depressive symptoms was identified, suggesting the need for psychological interventions combined with dietary intervention programs for vulnerable populations with poor diet satisfaction and increased difficulty with meal planning. Research should attempt to identify additional psychological risk factors associated with dietary self-management and its impact on dietary practices, fatigue, and diabetes health outcomes. Additionally, tailored intervention programs for older adults with diabetes who are susceptible to fatigue should be developed based on their functional and cognitive status to improve diet- and health-related quality of life and well-being.

## Figures and Tables

**Figure 1 ijerph-17-08823-f001:**
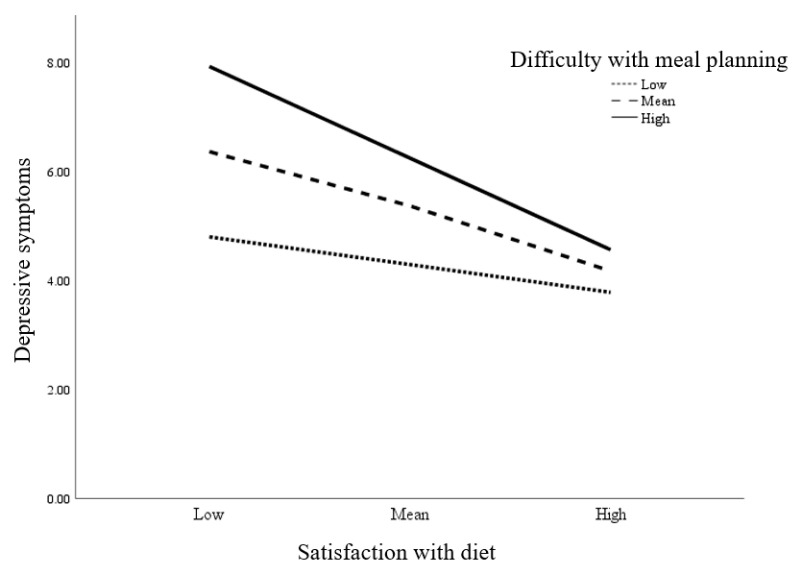
Moderating effect of difficulty with meal planning on the relationship between satisfaction with diet and depressive symptoms in older adults with diabetes. Low, mean, and high levels of satisfaction with diet and difficulty with meal planning were determined at −1SD from the mean (Low), mean, and +1SD from the mean (High).

**Figure 2 ijerph-17-08823-f002:**
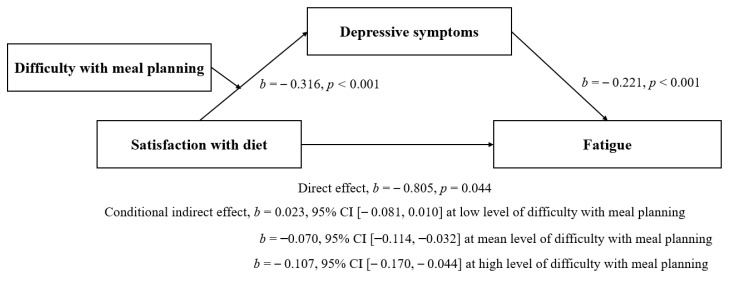
The moderated-mediation effect of difficulty with meal planning on the relationship between satisfaction with diet and fatigue through depressive symptoms in a sample of older adults with diabetes. Note: *b* = −0.147, *p* = 0.194 for low level of difficulty with meal planning; *b* = −0.316, *p* < 0.001 for mean of difficulty with meal planning; *b* = −0.484, *p* < 0.001 for high level of difficulty with meal planning.

**Table 1 ijerph-17-08823-t001:** The prevalence and severity of fatigue by sociodemographic, clinical, and psychological characters, and eating context in older adults with diabetes (*n* = 127).

Potential Predictors		Prevalence of Moderate or Severe Fatigue *n* (%)	Severity of Fatigue M ± SD	*p*
Sociodemographic Characteristics				
Age	<75 years	20 (32.3)	3.73 ± 1.79	0.428
	≥75 years	42 (67.7)	4.00 ± 1.81	
Gender	Men	27 (43.5)	3.88 ± 1.57	0.937
	Women	35 (56.5)	3.91 ± 1.94	
Marital Status	Widowed/divorced/separated	30 (48.4)	3.97 ± 1.84	0.685
	Married/partnered	32 (51.6)	3.84 ± 1.78	
Education	<High School Education	32 (51.6)	4.01 ± 2.15	0.476
	≥High School Education	30 (48.4)	3.78 ± 1.52	
Household Income	≤1,000,000 KRW/month	40 (64.5)	4.14 ± 1.88	0.295
	>1,000,000 KRW/month	22 (35.5)	3.78 ± 1.76	
Clinical Characteristics				
Years Having Diabetes	<10 years	27 (43.5)	3.82 ± 1.77	0.662
	≥10 years	35 (56.5)	3.97 ± 1.84	
Comorbidities	<2	43 (69.4)	3.98 ± 1.83	0.518
	≥2	19 (30.6)	3.76 ± 1.75	
Psychological Characteristics				
Difficulty with Meal Planning	Not at All Difficult	19 (30.6)	3.45 ± 1.83	0.012
	Not so Difficult or Somewhat Difficult	17 (27.4)	3.78 ± 1.59	
	Very Difficult or Extremely Difficult	26 (41.9)	4.56 ± 1.80	
Satisfaction with Diet	<Median	38 (61.3)	4.32 ± 1.74	0.004
	≥Median	24 (38.7)	3.41 ± 1.75	
Burden of Diet Therapy	<Median	34 (54.8)	4.00 ± 1.74	0.525
	≥Median	28 (45.2)	3.80 ± 1.87	
Perceived Merits of Diet Therapy	<Median	34 (54.8)	4.03 ± 1.81	0.387
	≥Median	28 (45.2)	3.75 ± 1.79	
Depressive symptoms	Not-depressed	24 (38.7)	3.40 ± 1.73	<0.001
	Mild Depression	23 (37.1)	3.62 ± 1.53	
	Severe Depression	15 (24.2)	5.41 ± 1.33	
Eating Context				
Exclusively Eating Alone ^†^	No	41 (66.1)	3.70 ± 1.73	0.037
	Yes	21 (33.9)	4.44 ± 1.89	
Eating out	Less than 2 times a week	48 (77.4)	4.11 ± 1.81	0.028
	3 times or more a week	14 (22.6)	3.32 ± 1.66	

Note: M, mean; SD, standard deviation; Korean Won (1000 KW ≈ 1.2 USD); ^†^ Eating every meal alone.

**Table 2 ijerph-17-08823-t002:** Multiple linear regression for predicting mediation effect of depressive symptoms on fatigue.

	Depressive Symptoms	Fatigue	
		Model 1	Model 2
Difficulty with Meal Planning	0.487 * [0.118, 0.855]	0.328 ** [0.136, 0.519]	0.232 * [0.049, 0.415]
Satisfaction with Diet	−0.292 ** [−0.453, −0.130]	−0.155 *** [−0.239, −0.071]	−0.098 * [−0.180, −0.016]
Burden of Dietary Restrictions	0.021 [−0.137, 0.180]	0.010 [−0.072, 0.093]	0.006 [−0.070, 0.083]
Perceived Value of Diet Therapy	−0.241 ** [−0.409, −0.072]	0.013 [−0.074, 0.101]	0.061 [−0.023, 0.145]
Eating Alone	1.209 [−0.085, 2.504]	0.541 [−0.131, 1.213]	0.303 [−0.331, 0.936]
Frequency of Eating Out	0.061 [−0.430, −0.551]	0.131 [−0.124, 0.386]	0.119 [−0.118, 0.356]
Depressive symptoms			0.197 *** [0.110, 0.284]
Adjusted *R*^2^	0.197	0.164	0.279
*R*^2^ change			0.094
*F*	6.449 ***	5.096 ***	6.986 ***
*F* change			15.476 ***
VIF	1.003–1.090	1.004–1.145	1.084–1.438
Durbin–Watson	1.718		1.180
*p*-value	<0.001	<0.001	<0.001

Note. VIF, variance inflation factor; * *p* < 0.05, ** *p* < 0.01, *** *p* < 0.001.

**Table 3 ijerph-17-08823-t003:** Moderating effect of difficulty with meal planning in the relationship between satisfaction with diets and depressive symptoms.

		Depressive Symptoms					
	*b*	*SE*(*b*)	*t*	*p*	95% CI (Lower, Upper)	△*R^2^*	*p*
Satisfaction with Diet	−0.316	0.084	−3.780	<0.001	−0.481, −0.150		
Difficulty with Meal Planning	0.621	0.184	3.376	0.001	0.25, 0.984		
Satisfaction with Diet * Difficulty with Meal Planning	−0.107	0.052	−2.058	0.042	−0.211, −0.004	0.028	<0.001
Moderated by Difficulty with Meal Planning *							
Low Difficulty with Meal Planning	−0.147	0.112	−1.306	0.194	−0.369, 0.076		
Mean Difficulty with Meal Planning	−0.316	0.083	−3.780	<0.001	−0.481, −0.150		
High Difficulty with Meal Planning	−0.484	0.122	−3.982	<0.001	−0.725, −0.244		

* Bootstrap results for indirect effect.

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
