# Peer review of "A Moderated-Mediation Model of the Relationship between Dietary Satisfaction and Fatigue in Older Adults with Diabetes: The Role of Meal Planning and Depressive Symptoms"

_ijerph, 2020, doi:10.3390/ijerph17238823_

Round 1

Reviewer 1 Report

The aim of the study is to examine the complex pathways between diet-related quality of life  with diabetes, depressive symptoms, and fatigue in older adults. More specifically, an analysis was conducted to examine the interrelationship between diet-related quality of life and fatigue severity and the mediating role of depressive symptoms in the relationship between factors of diet-related quality of life and fatigue. A convenience sample of adults aged 65 years or more (N = 127) with diabetes completed a cross- sectional survey including measures of fatigue severity, diet-related quality of life, and depressive symptoms. A moderated-mediation analysis was conducted to determine the relationship between these variables. Diet satisfaction was negatively related to fatigue, mediated by depressive  symptoms. In the moderated-mediation model, diet satisfaction had a conditional effect on fatigue through the mediating effect of depressive symptoms moderated by meal planning difficulty. The findings  supported the theory that fatigue was associated with decreased diet satisfaction, which was influenced by difficulties with meal planning in older adults with diabetes. Dissatisfaction with diet-related life and the psychological burdens of dietary management in older adults with diabetes appear to contribute to increased psychological distress such as depressive symptoms. The paper is quite interesting and well done. I have only a remark about to moderate the editing of english language and style of text.

Author Response

Dear Reviewer,

I thank you and the reviewers for your thoughtful suggestions and insights. The manuscript has benefited from these insightful suggestions.

The manuscript has been rechecked and the necessary changes have been made in accordance with your suggestions. The responses to all comments have been prepared and written in red font.

Thank you for your support.

Best regards,

Reviewer 2 Report

Overview:

The study investigated the relationship between dietary satisfaction and fatigue in diabetic adults aged 65 years old or older using a moderated-mediation model. The authors found that individuals with higher diet satisfaction had reduced prevalence of moderate or severe fatigue, mediated by depressive symptoms. In addition, perceived higher meal planning difficulty was associated with lower diet satisfaction and higher fatigue. The authors suggest that the findings demonstrate that social support, including support with meal planning practices, could positively improve self-care management of vulnerable populations with diabetes. Although the study is interesting and the authors have largely used appropriate tools for the investigation, some sections of the manuscript are difficult to follow and the rationale behind using multiple mediation models is absent. Furthermore, assessment of the criteria over a longer period of time (rather than only in the previous week) would provide a more accurate picture of the associations described. The authors should discuss this as a potential limitation of the current study. Further comments are listed below.

Introduction

-Sentence of lines 56-58: it is not obvious what the authors suggest to be a ‘predictor of fatigue’. Please revise the sentence to make this clearer.

Methods:

-Have the authors considered the effects of medication on fatigue and depressive symptoms? Other than antihyperglycemic agents were the subjects included in the study on any other medication? Authors should discuss the potential perceived burden of taking several different medication on individual’s quality of life and mood.

-On a similar note, did the individuals have any other underlying healthy condition apart from diabetes that could influence their quality of life?

-Line 93: what exactly do the authors mean by ‘living independently’? Does this mean the individuals do not require assistance, or live alone? In both cases, could this situation play a role in how they feel, linking to depressive symptoms?

-Line 103 and 104: should a comma replace one of the ‘and’?

-Line 122: same comment as above.

-Both fatigue and depressive symptoms were determine on how the participant felt over the last week. Could the authors comment on the potential impacts of measuring this over longer periods of time? Are the same findings expected to be observed if the participants were given the questionnaires on multiple occasions, same every week for 1 month?

-Were any of the volunteers included in the study supported by a dietitian?

Results:

-Line 168: Here the authors mention that fatigue was more severe in those who ate out frequently, however, the data presented in table 1 shows that prevalence of severe or moderate fatigue was lower in those who ate out 3 or more times a week. How have the authors come to this conclusion?

-The rationale behind using two models, rather than just one, to test the mediation effect of depressive symptoms on fatigue needs to be discussed. Also, the sensitivity of the models needs to be made clear.

-Table 3: very difficult to interpret this table and columns have been compressed too much

-Figure 1 and 2: are the values of B and b in these figures corresponding to the same thing (e.g. upper value or direct effect)? Please make this clear in figure legends.

-Line 248-249: This sentence is confusing, please re-phrase.

Discussion:

-Line 321: ‘population’ is used twice.

-Line 323: either ‘such as’ or ‘like’

-The authors may want to discuss the fact that findings of the study are representative of a ‘vulnerable population’ only. Other age groups, matching the criteria used in this study as much as possible (particularly living and eating alone or with others) should also be investigated in future studies.

-Author contributions are missing.

Author Response

Dear Reviewer,

I thank you for your thoughtful suggestions and insights. The manuscript has benefited from these insightful suggestions.

The manuscript has been rechecked and the necessary changes have been made in accordance with your suggestions. The responses to all comments have been prepared and written in red font.

Thank you again.

Best regards,

Heesook Son

Reviewer 3 Report

The article is relevant and contributes to public health.
However, the introduction needs to be clearer and more objective with a focus on the results found.
The type of research design and details of data collection are not presented.
In the regression models, the diagnostic tests of the model are not presented, nor if they reached the assumptions of the linear regression: normality of the residues, linearity, hemoscedasticity, absence of specification errors and absence of multicollinearity.

Author Response

(The authors gave the same response as above.)

Reviewer 4 Report

The manuscript has high relevance and it is well written. The results are well presented. I only suggest that the authors discuss a little more about why they have this difficulty with meal planning. Is it just because of  satisfaction of the diet?

Is the frustration of not being able to maintain meal planning related to depression or is depression the cause of the difficulty in following the planning?

Author Response

(The authors gave the same response as above.)
